# ART: rule bAsed futuRe-inference deducTion

**Mengze Li[1], Tianqi Zhao[1], Jionghao Bai[1], Baoyi He[1], Jiaxu Miao[1], Wei Ji[1], Zheqi Lv[1], Zhou Zhao[1*], Shengyu Zhang[1*], Wenqiao Zhang[1*], Fei Wu[2,3]**

[1]Zhejiang University [2]Shanghai Institute for Advanced Study of Zhejiang University
[3]Shanghai AI Laboratory

## Abstract

Deductive reasoning is a crucial cognitive ability of humanity, allowing us to derive valid conclusions from premises and observations. However, existing works mainly focus on language-based premises and generally neglect deductive reasoning from visual observations. In this work, we introduce rule b**A**sed futu**R**e-inference deduc**T**ion (**ART**), which aims at deducing the correct future event based on the visual phenomenon (a video) and the rule-based premises, along with an explanation of the reasoning process. To advance this field, we construct a large-scale densely annotated dataset (**Video-ART**), where the premises, future event candidates, the reasoning process explanation, and auxiliary commonsense knowledge (*e.g.*, actions and appearance) are annotated by annotators. Upon Video-ART, we develop a strong baseline named **ARTNet**. In essence, guided by commonsense knowledge, ARTNet learns to identify the target video character and perceives its visual clues related to the future event. Then, ARTNet rigorously applies the given premises to conduct reasoning from the identified information to future events, through a non-parametric rule reasoning network and a reasoning-path review module. Empirical studies validate the rationality of ARTNet in deductive reasoning upon visual observations and the effectiveness over existing works.

## 1 Introduction

Deductive reasoning is a systematic method that rigorously follows a set of explicitly given constraints (*i.e.*, rules) to deduce valid conclusions from empirical facts through logical inferences (Sanyal et al., 2022b). It represents a corner-

---

First author.
mengzeli@zju.edu.cn

*Corresponding author.
zhaozhou@zju.edu.cn
sy_zhang@zju.edu.cn
wenqiaozhang@zju.edu.cn

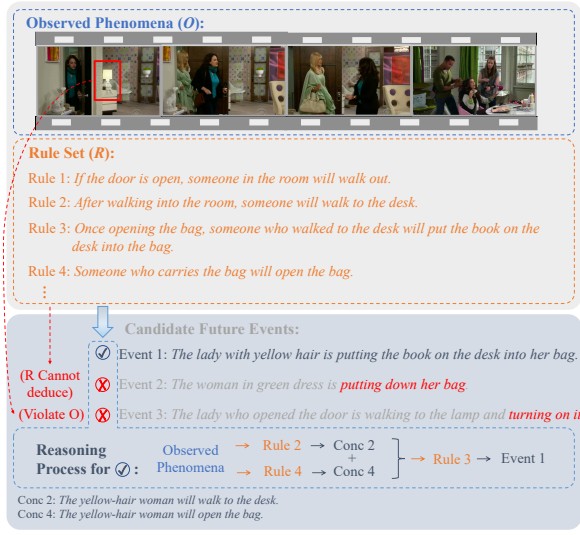

Figure 1: An illustration of the ART task. With the observed phenomena *O* and the rule set *R*, the ART task aims at reasoning out the correct one from candidate future events and explaining the reasoning process based on the rule set *R*.

stone of human psychological functioning, serving as an indispensable aspect of our daily cognitive processes. For example, human beings possess the capability to utilize the given rule set (*R*) to deduce future events (*F*) through the interpretation of observed phenomena (*O*). To illustrate:

- Supposing that the rule *R: as the self-protection, the person will release the hot object, once burned.* holds, and we observe *O: a man holds a very hot teacup,* it follows logically that we anticipate *F: he will release the cup.*

- Under the premise that *R: after getting home, my dad will definitely smoke to relieve anxiety;*, the observation *O: dad returns home from work at night,* should lead to the future event *F: he will smoke*.

Despite deductive reasoning being acknowledged

as a fundamental cognitive competency of humanity (Rips, 1994; Evans et al., 1993), there is a scarcity of investigations in designing AI systems that are capable of executing deductive reasoning in the multi-modal field.

To advance the research, we simulate the deductive reasoning of human beings and propose a rule b**A**sed futu**R**e-inference deduc**T**ion task (**ART**). Overall, in aligning with the established deductive reasoning studies within the NLP community (Sanyal et al., 2022b), ART should select the correct textual *future events* (correct conclusions) from potential candidates, rigorously based on the given language rule set and the observed phenomena. As depicted in Figure 1, the *observations* in ART are illustrated by videos, which is grounded in the fact that visual information has a profound impact on the human brain, accounting for an impressive 83% of all inputs (Rosenblum, 2011). Upon the video observations, ART reasons on a crucial ingredient of videos, *i.e.*, human actions, and endeavors to derive the most accurate future event, constrained by the given rule set and the video observation. In addition to deducing future events, ART should provide explanations of the reasoning process by generating the rule chain. In formal terms, our ART task is close to the well-established video-language inference task (Liu et al., 2020; Li et al., 2021), which assesses the accuracy of language descriptions in relation to input videos. Analyzing the difference between the two tasks can facilitate the advancement of our new field. Compared with the video-language inference, our ART task has the following characteristics: (1) The task demands a transparent explanation of the reasoning process; (2) In accordance with deductive reasoning (Sanyal et al., 2022b), the reasoning processes should be constrained by the rule sets (regardless of the perceived irrationality of provided rules) that are explicitly assigned to each individual sample, while relying on implicitly learned rule constraints that might come from other samples could cause mistakes.

To promote multi-modal deductive reasoning research and meet the demands of the ART task, we introduce a new dataset, named **Video-ART**, consisting of $23,895$ samples. Careful annotation was performed by annotators and verifiers with strong logical reasoning skills, who mainly focused on two key aspects: (1) They targeted to design the rule sets and the candidate future events that are closely associated with the visual information presented; (2) The annotators provided the correct future events and a rule-based explanation of the reasoning process. In addition, to enhance the AI system's deduction from visual semantics in highly unstructured videos, which are composed of densely arranged pixels, we have carefully annotated the commonsense knowledge of the target objects, including their appearance and related actions.

To lay the groundwork for future research, we propose a strong baseline for the rule based future-inference deduction, named **ARTNet**. ARTNet mainly consists of three components, *i.e.*, knowledge-guided target perception (KTP), non-parametric rule reasoning network (RRN), and reasoning path review (RPR). **KTP** learns to identify the target character and corresponding visual clues related to the upcoming event through multi-task learning and commonsense knowledge annotations such as actions and appearance. Inspired by traditional graph-theoretic algorithms, **RRN** performs layer-by-layer reasoning through a purpose-built non-parametric rule reasoning network, uncovering the reasoning paths from the identified visual clues to potential future events. RRN offers two advantages for the ART task over traditional models: (1) RRN provides explanations of its rule-based reasoning process. (2) RRN avoids rote memorization of rules within the training data and ensures the rigorous application of the sample-specific rule set. Furthermore, the **RPR** module validates the semantic consistency between the rule reasoning paths uncovered by RRN, the video observations, and the future event descriptions.

Overall, the main contributions of this work are three-fold:

- We propose the rule based future-inference deduction task, through imitating human cognition. To the best of our knowledge, this is an early exploration of deductive reasoning in the multi-modal domain.

- We construct a large-scale dataset Video-ART [1] to promote multi-modal deductive reasoning research. Video-ART consists of $23,895$ examples where dense annotations including the rule set, reasoning processes, and auxiliary commonsense knowledge are provided.

---

[1]Please contact mengzeli@zju.edu.cn for dataset acquisition.

- We contribute a strong baseline, ARTNet, tailored for the ART task. Experimental results on the Video-ART dataset validate the effectiveness of ARTNet over the state-of-the-arts.

## 2 Related Work

**Video-Language Inference.** As the development of the deep learning (Wu et al., 2020; Miao et al., 2021; Wu et al., 2022; Ji et al., 2023b,c), the visual and language related tasks attracts more and more attentions (Wu et al., 2023; Li et al., 2023a; Miao et al., 2022; Ji et al., 2023a; Li et al., 2022b,c). The video-language inference task aims at judging the correctness of the textual conclusion, based on the video information and the language description (Li et al., 2020; Tang et al., 2021; Chen and Kong, 2021; Zhang et al., 2019; Gokhale et al., 2022). (Liu et al., 2020) proposes a carefully labeled dataset for this task and introduces a strong baseline to further develop this field. Based on this dataset, (Li et al., 2021) designs a new model based on the graph network and validate it.

**Future Event Prediction.** Our task is formally related to future event prediction, whose goal is to infer the future from known facts (Surís et al., 2021; Vondrick et al., 2016; Epstein and Vondrick, 2021). This field encompasses many tasks, such as the future trajectory prediction (Kim et al., 2021; Chen et al., 2022; Li et al., 2022a; Alahi et al., 2016), the action prediction (Abu Farha et al., 2018; Kitani et al., 2012; Lan et al., 2014; Ryoo, 2011), and future object localization (Jia et al., 2022; Peri et al., 2022). Graph-based methods (Zhang et al., 2021, 2022) are often considered alternative solutions for such tasks.

**Deductive Reasoning.** Reasoning is an important skill for human beings to understand the world (Byrne, 1989; Rips, 1994) To promote the development of human-like reasoning AI systems, many researchers have invested in this field (Ebrahimi et al., 2021; Li et al., 2023b; Calimeri et al., 2021; Sanyal et al., 2022a). (Sanyal et al., 2022b) proposes a deductive reasoning task in the field of NLP and presents a powerful model that outperforms previous methods.

## 3 Deductive Reasoning Dataset

Our rule b**A**sed futu**R**e-inference deduc**T**ion task (**ART**) requires the AI system to (1) select the correct future event from the candidate events by reasoning on the rule set and the observation (a

| Category | Subcategory |
|---|---|
| Appearance | Gender, Hair Length, Age |
| Clothing | Length of Lower-body Clothing, Type of Lower-body Clothing, Type of Upper-body Clothing, Sleeve Length, 3 Other Outfits, 9 Colors of Upper-body Clothing, 9 Colors of Lower-body Clothing |
| Action | Intransitive Verb, Transitive Verb, Object |
| Sentiment | No Subcategories |
| Scene | No Subcategories |

Table 1: Statistics of commonsense knowledge types for Video-ART.

video); (2) explain the logical chain leading to the final conclusion based on the rules. Taking into account the ART task characteristics, we propose an exhaustively labeled large-scale dataset, named **Video-ART**.

### 3.1 Data Collection

We collect the videos in our dataset from two sources: (1) Parts of the video clips are manually intercepted from 80 American movies, including Broke Girls, Grey's Anatomy, Mr. Bean, etc. These videos are of high quality, with rich character actions and emotions, and rigorous plot logic. (2) Other videos are carefully selected from the existing datasets, Charades (Sigurdsson et al., 2016) and TO-MAR (Li et al., 2023b), both of which are the human-center datasets. These videos consist of many different actors and scenes. The appearance, clothing, etc of the characters are richer.

Both sources of data have their own characteristics and combined together may provide a relatively comprehensive testbed for the ART task. Some collected videos are not suitable for our ART task, such as videos with few actions or blurred videos in which key details cannot be clearly distinguished.

### 3.2 Data Annotation

With the collected videos, we rigorously design the ART task examples for each data and manually validate all examples. In addition, we annotate the commonsense knowledge for all video characteristics in detail to assist AI system training, including human appearance, clothing, actions, semantics, and scenes located.

**Commonsense Knowledge Annotation.** The

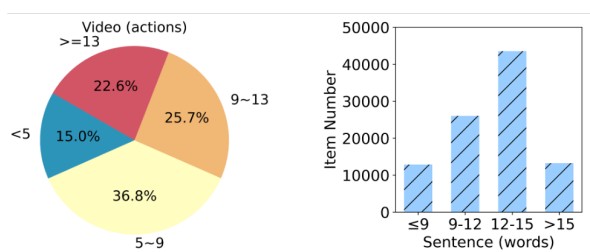

Figure 2: Statistics about the dataset: (1) the videos with different action numbers; (2) the future events with different word numbers.

annotated categories and subcategories of commonsense knowledge for each video characteristic are shown in Table 1. More annotation details are shown in the appendix.

**ART Task Annotation.** Our annotations for the ART task contain two aspects: (1) the rule set and the candidate future events for each video; (2) the labels indicating the correct future event and the complete rule chain as the explanation coming for the correct event. Specifically, to achieve rigorous labeling, 4 doctoral and undergraduate students from the top 50 universities in the world are responsible for annotation. Firstly, the annotators crop out appropriate video clips as video observation. In reference to the commonsense knowledge and the subsequent video content of the cropped video clips, the correct future event and the rule-based explanation of the reasoning process for this event (the rule chain) are annotated. Then, according to the correct event, the confusing items in both the candidate future events and the rule set are supplemented.

**Validation.** The verifiers with strong logical abilities are responsible for verifying the labeled examples. The examples not agreed by them are relabeled or discarded.

### 3.3 Dataset Analysis

Our dataset has the following characteristics: **(1) Deductive Reasoning Orientated.** The Video-ART dataset is strictly designed according to the ART task. It is the first deductive reasoning dataset in the multi-modal field collected from different human scenes. **(2) Large-scale.** The dataset consists of $23,895$ examples. Among them, $5,922$ examples are collected from movies and $17,973$ examples come from real-life scenarios. **(3) Diversity.** (1) The examples in the dataset are rich in scenarios, including residence, hospital, restaurant, etc. (2) The dataset covers a variety of activities

such as working, cleaning, cooking, etc.

In addition, on average there are $4$ future events in each example. The average length of the videos is $24.5$ seconds. Detailed statistics are shown in Figure 2. We show more statistical results in the appendix.

## 4 Method

We propose a new task, rule b**A**sed futu**R**e-inference deduc**T**ion (**ART**), and design a targeted model named **ARTNet**. According to the task characteristics, we contribute the non-parametric rule reasoning module for ARTNet. In addition to the key reasoning module, the knowledge-guided perception module and the rechecking module are introduced to assist in the completion of the ART task.

**Task Formulation.** Given an observation (a video) $\mathcal{V}$, a rule set (multiple rules) $\mathcal{S} = \{\mathcal{S}_i\}_{i=1}^{N_{\mathcal{S}}}$, and candidate future events $\mathcal{C} = \{\mathcal{C}_i\}_{i=1}^{N_{\mathcal{C}}}$ described by the natural language, the ART task aims to reason out the correct future event and explain the reasoning process based on the rules. We define the model with the parameter $\Theta$ for the ART task as $\mathcal{M}$. Then, the training optimization function $\delta(.)$ of $\mathcal{M}$ is represented as:

$$
\begin{aligned}
&\delta(\mathcal{C}, \mathcal{S}, \mathcal{V}; \Theta) \\
&\quad = \max_{\Theta} \xi(\epsilon(\mathcal{C}, \mathcal{S}, \mathcal{V}), \mathcal{M}(\mathcal{C}, \mathcal{S}, \mathcal{V}; \Theta)),
\end{aligned} \quad (1)
$$

where $\Theta$ is a learnable parameter. The function $\epsilon(.)$ generates the ground truth and the function $\mathcal{M}(.)$ outputs the model prediction. The function $\xi(.)$ calculates the consistency of $\epsilon(.)$ and $\mathcal{M}(.)$.

**Rule Transformation.** Before describing our ARTNet structure, we shed light on the intriguing transformation of rules within the ART task. The ART task revolves around predicting future events based on observed information, employing rules as the means of inference. These rules can be perceived as an intricate mapping of crucial information bridging two consecutive events. It is important to note that the core driver of event progression for the target character lies in the changes in actions. Hence, in our proposed baseline, we adopt an approximation where rules are represented as mappings of actions. We employ the robust StanfordNLP toolkit (Manning et al., 2014) to identify the key actions within each rule, which serve as the basis for future-inference deduction and explanation.

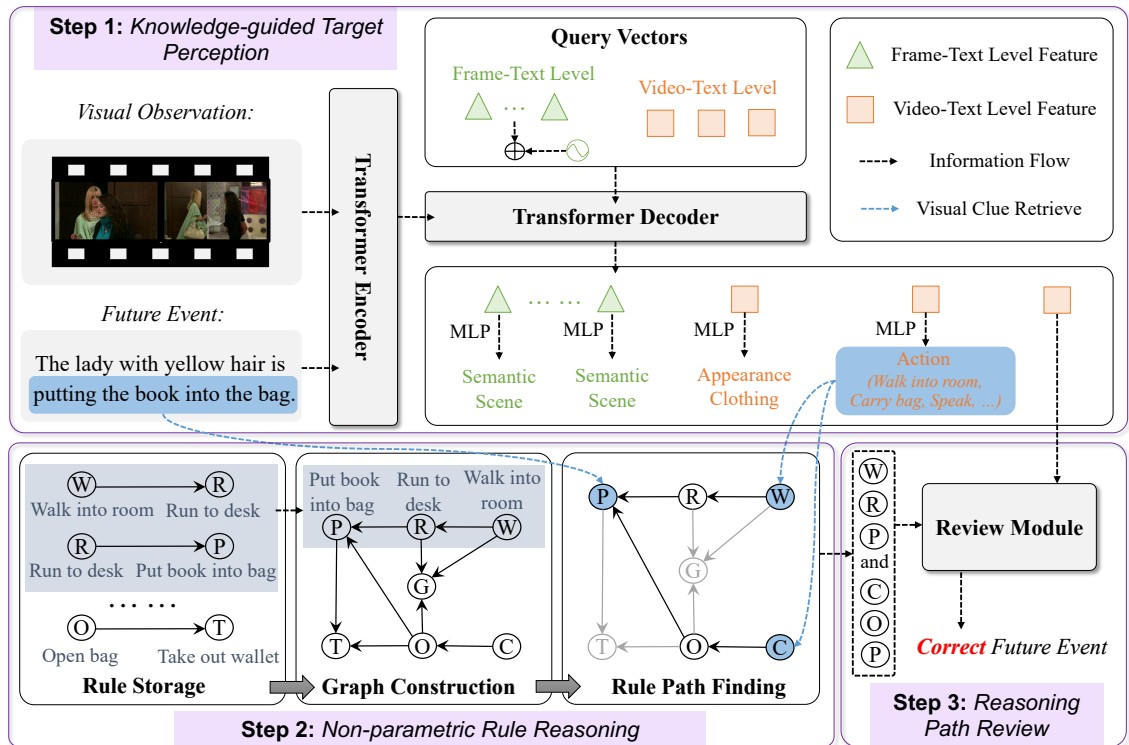

Figure 3: Overview of our ARTNet model for the ART task. It consists of three steps. Step 1: Knowledge-guided Target Perception, which focuses on the language-described person in the video and identifies her key actions. Step 2: Non-parametric Rule Reasoning, which constructs the rule graph (action graph), and finds the connected action path between the video actions and the future-event action with Dijkstra's algorithm. Step 3: Reasoning Path Review, which finally checks whether the future event, the video, and the found rule paths match to determine the correctness of the future event.

**Model Pipeline.** As shown in Figure 3, our ARTNet judges the correctness of candidate future events $\mathcal{C}$ one by one based on the given rule set $\mathcal{S}$ and the video $\mathcal{V}$. It chooses the correct item with the highest probability as the prediction result and outputs the rule-relied inference process as an explanation. Specifically, the whole process is divided into three steps: **Step 1 (Section 4.1):** By analyzing the input video $\mathcal{V}$ and the future event $\mathcal{F}$, the knowledge-guided target perception module focuses on the person described by the language event $\mathcal{F}$ and predicts her key visual action. **Step 2 (Section 4.2)** The non-parametric rule reasoning network constructs the rule graph based on the action chains stored in the rule memory. Then, it finds the connected rule paths between the future event action and the visual action from the video $\mathcal{V}$. If no path is found, we judge the future event $\mathcal{F}$ is wrong. **Step 3 (Appendix):** The review module of ARTNet reasons on the found connected rule paths and the cross-modal feature containing the semantics of the video $\mathcal{V}$ and the future event $\mathcal{F}$. The module outputs the correct probability $p_e$ of the

future event $\mathcal{F}$. We choose the one with the highest probability from the candidate future events as the final prediction result of the deductive reasoning task, ART. The corresponding rule path in Step 2 is viewed as the explanation for the prediction result. Step 3 is introduced in the appendix in detail.

## 4.1 Knowledge-guided Target Perception

Identifying the target character and the corresponding action knowledge related to the natural language future event $\mathcal{F}$ from the input video $\mathcal{V}$ is an essential step before the reasoning based on the rule set $\mathcal{S}$ (which is transformed into action chains in the preprocessing). Toward this target, we leverage human-annotated commonsense knowledge labels (human appearance, clothing, semantics, scene, and actions) of the target video person described by the textual future event $\mathcal{F}$ to train the model with transformer-based multi-task learning.

Specifically, the knowledge-guided visual perception module is designed based on the transformer architecture. The transformer encoder extracts the cross-modal feature $\mathbf{F}_c$ from the video

$\mathcal{V}$ and the future event $\mathcal{F}$. We define two types of query vectors to analyze the cross-modal feature $\mathbf{F}_c$ with the transformer decoder. It includes the frame-text level queries $\mathbf{Q}_f = \{\mathbf{q}_f^i\}_{i=1}^{N_f}$ and the video-text level queries $\mathbf{Q}_v = \{\mathbf{q}_v^i\}_{i=1}^3$ used to analyze the cross-modal semantics, where $N_f$ is the number of frames. The transformer decoder distinguishes different types of query vectors according to the injected type embeddings. Then, it reasons the corresponding features (frame-text level features $\mathbf{F}_f = \{\mathbf{f}_f^i\}_{i=1}^{N_f}$ and video-text level features $\mathbf{F}_v = \{\mathbf{f}_v^i\}_{i=1}^3$) relying on the query vectors ($\mathbf{Q}_f$ and $\mathbf{Q}_v$). By analyzing the resulting features ($\mathbf{F}_f$ and $\mathbf{F}_v$), we predict all commonsense knowledge. We take the action knowledge predcition as an example, the others are shown in appendix.

**Action Knowledge Prediction.** For action knowledge, we do not predict it frame-by-frame like sentiments and scenes. It is because there are multiple actions for the target person in each frame and the frame-by-frame prediction introduces too much burden to ARTNet, which leads to difficult model training. Therefore, the model directly counts the actions contained in the video rather than each frame. The process of judging whether the i-th action exists in the video $\mathcal{V}$ with the video-level query $\mathbf{f}_v^1$ is represented as:

$$\mathbf{p}_{ac}^i = \mathrm{softmax}(MLP_{ac}^i(\mathbf{f}_v^1)), \qquad (2)$$

where the $MLP_{ac}^i$ is the MLP applied specifically for the i-th action prediction. The $\mathbf{p}_{ac}^i$ is the probability of action existence or not.

## 4.2 Non-parametric Rule Reasoning

The rule-based reasoning of the ART task requires the model not to memorize the rules in the training set and has strong interpretability. Towards this end, we propose the non-parametric rule reasoning network, based on the traditional graph theory rather than the neural network.

In detail, the action set contained in the rule set (action chain set) is represented as $\mathcal{A} = \{\mathcal{A}_i\}_{i=1}^{N_{\mathcal{A}}}$. As shown in the the Figure 3, the non-parametric rule reasoning network contains three steps: **(1) Action Chain Storage.** In order to facilitate subsequent processing, we *split* the action chains into multiple single-step relational maps and store them in the memory. Taking the action chain $\mathcal{A}_1 \to \mathcal{A}_2... \to \mathcal{A}_r$ as an example, the splitting

process is represented as:

$$(\mathcal{A}_1 \to \mathcal{A}_2),...,(\mathcal{A}_{r-1} \to \mathcal{A}_r)$$
$$= split(\mathcal{A}_1... \to \mathcal{A}_r). \qquad (3)$$

Notably, for the action chain form of combinatorial inference in the action chain set $\mathcal{A}_i + \mathcal{A}_j \to \mathcal{A}_k$, we store two single-step relational maps, $\mathcal{A}_i \to \mathcal{A}_k$ and $\mathcal{A}_j \to \mathcal{A}_k$. **(2) Graph Construction.** We construct the action graph $\mathcal{G}(\mathcal{A}, \mathcal{U})$ by *connecting* all the single-step relational maps $(\mathcal{A}_i \to \mathcal{A}_{i+1}),...,(\mathcal{A}_{i+n-1} \to \mathcal{A}_{i+n})$ in the memory, where $\mathcal{U}$ represents the edges between the actions $\mathcal{A}$ in the graph. The construction process is formalized as:

$$\mathcal{G}(\mathcal{A}, \mathcal{U}) = connect((\mathcal{A}_i \to \mathcal{A}_{i+1}),...,$$
$$(\mathcal{A}_{i+n-1} \to \mathcal{A}_{i+n})). \qquad (4)$$

**(3) Action Path Finding.** Firstly, we need to find the starting nodes $\{\mathcal{A}_{s_i}\}_{i=1}^{N_{\mathcal{A}_s}}$ and the ending node $\mathcal{A}_e$ of the target action paths in the constructed graph $\mathcal{G}(\mathcal{A}, \mathcal{U})$. The starting nodes $\{\mathcal{A}_{s_i}\}_{i=1}^{N_{\mathcal{A}_s}}$ are determined by matching the actions predicted in step 1 (Section 4.1) and each graph node (action). Similarly, we find the ending node $\mathcal{A}_e$ by matching each graph node (action) and the action of the future event $\mathcal{F}$ detected by the widely used tool, StanfordNLP (Manning et al., 2014). Secondly, we find all the connected action paths between the starting nodes $\{\mathcal{A}_{s_i}\}_{i=1}^{N_{\mathcal{A}_s}}$ and the ending node $\mathcal{A}_e$ with Dijkstra's algorithm. Using the starting node $\mathcal{A}_{s_j}$ and the ending node $\mathcal{A}_e$ as an example, the path-finding process is:

$$\mathcal{A}_{s_j}... \to \mathcal{A}_e = Dijkstra(\mathcal{A}_{s_j}, \mathcal{A}_e, \mathcal{G}(\mathcal{A}, \mathcal{U})). \qquad (5)$$

Finally, we review all the found action paths again for violations of the action chains in the action chain set $\mathcal{S}$ and delete them. Notably, for the rule paths (like $\mathcal{A}_p \to \mathcal{A}_i \to \mathcal{A}_k$ and $\mathcal{A}_q \to \mathcal{A}_j \to \mathcal{A}_k$) involving combinatorial inference rules (like $\mathcal{A}_i + \mathcal{A}_j \to \mathcal{A}_k$), these action paths need to be merged into one and then checked. After merging, checking, and deleting, multiple action paths may be preserved. They need to be further verified in the next review module.

## 5 Experiments

We experiment with our **ARTNet** model on our proposed **Video-ART** dataset to verify the

model's effectiveness for the rule b**A**sed futu**R**e-inference deduc**T**ion task (**ART**). All experimental environments are deployed in Hikvision (https://www.hikvision.com/en/).

**Dataset.** The Video-ART dataset consists of data from real life scenes and movie scenes, which are randomly divided into $14,029/706/3,238$ (train/val/test) and $3,902/349/1,671$ (train/val/test), respectively. As stated in Section 3.1, both types of data have their own characteristics. To comprehensively evaluate the performance of the models, we conduct experiments in both scenarios.

**Evaluation Metrics.** Following previous deductive reasoning tasks (Sanyal et al., 2022b), our ART task requires that the correct future event in the candidate set and its explanation for the reasoning process are both unique. Therefore, we use "accuracy" to measure the correctness of the model predictions for them. The accuracy of the future event prediction and the explanation prediction are denoted as "Event_ACC" and "Exp_ACC", respectively. We consider an explanation prediction to be potentially correct only when future events are predicted accurately.

**Baselines.** Previous methods from other tasks cannot adopt our ART task in a direct manner. Thus, several state-of-the-art multi-modal and reasoning models are extended as the baselines to compare. Specifically, to make a comprehensive comparison, we take into account the following methods: (1) video-language inference methods: LF-VILA (Sun et al.), AHGN_SCL (Li et al., 2021), VIOLINet (Liu et al., 2020); (2) deductive reasoning methods: FAIRR (Sanyal et al., 2022b).

## 5.1 Performance Comparison

**Comparison with State-of-the-arts.** Our ARTNet model is compared with the baselines on the Video-ART dataset for the ART task. The experiment results are shown in Table 2. From the table, there are the following findings. **(A)** Compared with the baselines, our ARTNet model performs best and improves the accuracy by more than 4 points on all metrics. We contribute the improvement to 1) the commonsense knowledge guidance, which makes ARTNet focus on the target person and correctly identify the key actions from the video according to the future event; 2) the non-parametric rule reasoning, which implements layer-by-layer reasoning strictly according to the given rules and provides

| # | Methods | Trans. | *Event_ACC* | *Exp_ACC* |
|---|---------|--------|-----------|---------|
| | | Real Life Scene | | |
| 1 | **VIOLINet** (Yang et al., 2022) | | 29.2 | 22.9 |
| 2 | **AHGN_SCL** (Li et al., 2021) | | 31.3 | 27.1 |
| 3 | **FAIRR** (Sanyal et al., 2022b) | ✓ | 35.8 | 30.1 |
| 4 | **LF-VILA** (Sun et al.) | ✓ | 37.9 | 34.5 |
| 5 | **ARTNet** (Ours) | ✓ | 42.3 | 41.0 |
| | | Movie Scene | | |
| 1 | **VIOLINet** (Yang et al., 2022) | | 30.1 | 24.6 |
| 2 | **AHGN_SCL** (Li et al., 2021) | | 32.7 | 26.2 |
| 3 | **FAIRR** (Sanyal et al., 2022b) | ✓ | 36.2 | 32.5 |
| 4 | **LF-VILA** (Sun et al.) | ✓ | 37.8 | 35.6 |
| 5 | **ARTNet** (Ours) | ✓ | 42.5 | 42.0 |

Table 2: Comparison results between ARTNet and the state-of-the-arts. "Trans." indicates the transformer-based architecture.

| # | Methods | Trans. | *Event_ACC* | *Exp_ACC* |
|---|---------|--------|-----------|---------|
| | | Real Life Scene + Movie Scene | | |
| 1 | **VIOLINet** (Yang et al., 2022) | | 30.4 | 26.7 |
| 2 | **AHGN_SCL** (Li et al., 2021) | | 33.5 | 28.6 |
| 3 | **FAIRR** (Sanyal et al., 2022b) | ✓ | 35.7 | 32.1 |
| 4 | **LF-VILA** (Sun et al.) | ✓ | 38.2 | 35.8 |
| 6 | **ARTNet** (Ours) | ✓ | 42.5 | 41.8 |

Table 3: Comparison results between ARTNet and the state-of-the-arts on the merged dataset of the two scenarios. "Trans." indicates the transformer-based architecture.

the explanation for the reasoning process. **(B)** The baselines are extended from other tasks and lack targeted domain knowledge of our ART task, which leads to unsatisfactory performance.

**Comparison with baselines on the merged dataset.** We are interested in the ARTNet model performance on the whole dataset. Thus, we merge the two dataset parts, including the real-life examples and the movie examples, and experiment on them. The results of the baseline comparison are shown in Table 3. From the table, we have the following findings: **(A)** Compared with other baselines, our ARTNet model performs best. This once again demonstrates the rationality of two task-targeted modules, the Knowledge-guided Target Perception, and the Non-parametric Rule Reasoning, in the ARTNet model. **(B)** Training on the merged dataset of the two scenarios does not significantly improve the model performance, compared with being trained on a single scenario. It means the model cannot effectively transfer the learned

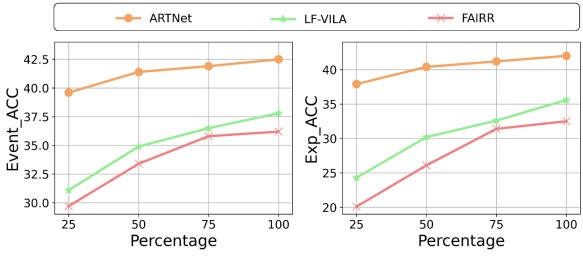

Figure 4: Comparison with the state-of-the-arts on different proportions of training set in the movie scene.

| CKG | RRN | Real Life Scene | | Movie Scene | |
|---|---|---|---|---|---|
| | | Event_ACC | Exp_ACC | Event_ACC | Exp_ACC |
| | | 37.7 | 34.9 | 38.1 | 35.1 |
| ✓ | | 39.5 | 37.8 | 40.5 | 38.3 |
| | ✓ | 40.7 | 39.5 | 41.4 | 40.8 |
| ✓ | ✓ | 42.3 | 41.0 | 42.5 | 42.0 |

Table 4: Ablation study of ARTNet on the Video-ART dataset. **CKG** is the **C**ommonsense-**K**nowledge **G**uidance, and **RRN** represents the non-parametric **R**ule **R**easoning **N**etwork.

knowledge between the two scenarios, which is due to the significant differences between the two scenarios: (1) Movie videos have a higher resolution, while real-life videos are limited by the lower resolution of the shooting devices. This results in the difference in the model's visual perception of the two scenarios. (2) Movie scenes have richer and more exaggerated plotlines, while real-life scenes have simpler plotlines. Thus, there are significant differences in the event reasoning of the two scenarios.

**Comparison with baselines on different training data volumes.** To evaluate the performance of the ARTNet model trained on different data volumes, we randomly select 25%, 50%, and 75% of the training data in the movie scene for experiments. The comparison results between ARTNet and baselines are shown in Figure 4. From it, we can observe that the accuracy of ARTNet is still higher than the state-of-the-arts with the low training data volume. We attribute it to the effectiveness of the knowledge-guided auxiliary training for the transformer module and the non-parametric rule reasoning network independent of the training data, which guarantees the model performance with less training data.

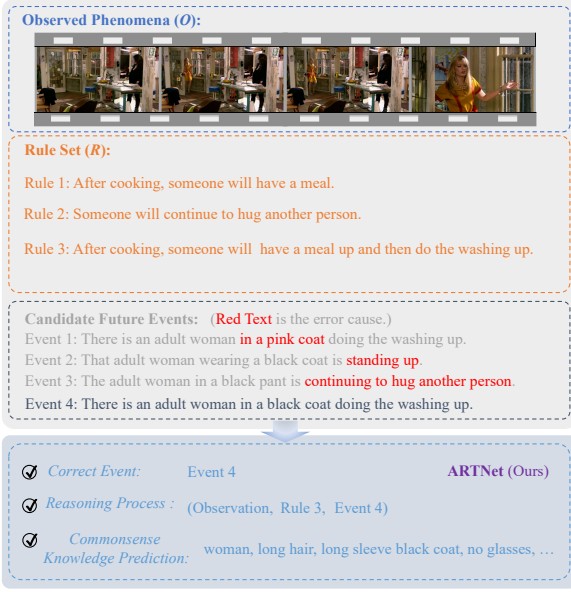

Figure 5: Case study of the ARTNet performance.

## 5.2 In-depth Analysis

**Ablation Study** We are interested in the contribution of each key module in our ARTNet model and design the ablation study. Specifically, we surgically remove the **C**ommonsense-**K**nowledge **G**uidance (**CKG**) and the non-parametric **R**ule **R**easoning **N**etwork (**RRN**) from our ARTNet model and get different architectures. Without RRN, the ARTNet model totally losses the rules exploit capabilities, which is necessary for the ART task. Thus, we replace the RRN module with the advanced NLP model, transformer (Vaswani et al., 2017), rather than simply removing it. The experimental results of the ablation study are shown in Table 4. According to the results, there are several findings: **(A)** After removing any key modules, the model performs worse. It proves that CKG can improve the model's ability to perceive key video information and RRN is able to achieve rigorous reasoning. Thus, both of them are indispensable. **(B)** When we replace RRN with transformer, the transformer model remembers the rule sets of the examples in the training set during training. This may cause the model to fail to strictly utilize the rule sets given by the test examples during inference, resulting in a decrease in accuracy. **(C)** It is more efficient to use key modules together than to use them separately. This demonstrates that the ART task completion can be significantly improved by combining CKG's information perception ability with RRN's knowledge reasoning ability.

**Case Study** To further demonstrate the effective-

ness of our ARTNet model in visual, we carefully design the case study. Specifically, we select an example from the Video-ART dataset and show the experiment results in Figure 5. From the figure, we can observe that our ARTNet model predictions are completely accurate. This intuitively demonstrates that ARTNet can achieve precise perception and rigorous reasoning for the ART task. In addition, we compare our ARTNet model with the state-of-the-art, and the ablation base model. These examples are shown in the appendix.

## 6 Conclusion

We study the deductive reasoning process in humans and propose a video-text deductive reasoning task, ART, which is an early exploration of deductive reasoning in the field of multi-modal. To promote this new task development, we propose a new dataset, Video-ART, and a strong baseline called ARTNet. Experiments prove the ARTNet effectiveness.

## Limitations

We propose a strong baseline, ARTNet, for the ART task, as a field foundation. The ARTNet baseline is limited to approximate the rules as the action chains to further process. In the future, we will update the ARTNet to improve the design of this part. We hope our work could promote the development of the multi-modal deductive reasoning.

## Acknowledgments

Our research is funded in part by the Key Project of the National Natural Science Foundation (No.62037001), and the Starry Night Science Fund at Shanghai Institute for Advanced Study (Zhejiang University).

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
