# OpenReview forum: "ART: rule bAsed futuRe-inference deducTion"
_EMNLP/2023/Conference — EMNLP 2023 Main_

### Official Review · Reviewer_urk3 · 2023-08-02

**Soundness:** 4

**Excitement:**

4: Strong: This paper deepens the understanding of some phenomenon or lowers the barriers to an existing research direction.

**Paper Topic And Main Contributions:**

This paper proposed a  rule based future-inference deduction task to explore the deductive reasoning in multi-modal domain.
Moreover, this paper construct a large-scale dataset to promote multi-modal deductive reasoning research, which consists of 23, 895 examples where dense annotations including the rule set, reasoning processes, and auxiliary commonsense knowledge are provided.
Finally, the author contribute a strong baseline, ARTNet, tailored for the ART task.

**Questions For The Authors:**

The over claim towards ``the first'' is not a big deal.
My main concerns is the real-world application, can you provide a more applicable usage towards your model and task. For example, whether the rule set can be generated by LLMs? And how is the performance of your method when facing generated rule set? Moreover, whether your dataset can be further expanded by LLMs in the aforementioned approach?

**Reasons To Accept:**

1. This work propose a different deduction reasoning in multi-modal domain.
2. This work contribute a meaningful dataset to support this task.
3. Experimental results shows that the deduction reasoning can be finished in this way.

**Reasons To Reject:**

1. The task proposed in this paper cannot be applied in real application, since there are no extra annotation, like the rule set.
2. This work is not the first to explore deductive reasoning in the multi-modal domain, please refer to compositional reasoning in VideoQA and ImageQA for more information. These task are not same as your task, but they explore programs to model the deductive reasoning in the multi-modal domain.

**Reproducibility:**

4: Could mostly reproduce the results, but there may be some variation because of sample variance or minor variations in their interpretation of the protocol or method.

**Reviewer Confidence:**

4: Quite sure. I tried to check the important points carefully. It's unlikely, though conceivable, that I missed something that should affect my ratings.

---

> ### Author Rebuttal · Authors · 2023-08-29
>
> ## Rebuttal to Reviewer urk3
>
> Thank you for your appreciation and patient feedback on this research paper. We will carefully respond to each of the comments one by one.
>
> ## 1. The task proposed in this paper cannot be applied in real application, since there are no extra annotation, like the rule set.
>
> We understand your concerns regarding the difficulty of obtaining the rule set in the real applications. We will clarify how to obtain the rule set and add a more complete and detailed description to the next paper version.
>
> The rule set in the real-life application can be viewed as a knowledge base. In real-world scenarios, there are several situations involved in obtaining the rule set:
>
> ***The rule set is already available.***
> In many fields, like the judicial/legal scenario, the knowledge base has been well-defined by the experts. After simple filtering and modifications, the rule sets can be built using the existing knowledge base. With the constructed rule sets, the deductive reasoning AI system may assist in completing prediction tasks in the given scenarios, such as judicial rulings.
>
> ***The rule set will be defined by humans.***
> In many critical fields (like counter-terrorism), human needs to participate in the construction of foundational rules. Specifically, in the field of security, humans are required to define different rule sets for various real-world scenes. The AI system can provide crime alerts based on video footage and the rule set specific to this scene.
>
> Furthermore, using LLMs is a good choice to reduce the manual labor cost of annotating rule sets in these fields. Specifically, the well-trained LLMs provide the base rule sets. The human annotators need to make simple corrections or adjustments to the generated rule sets.
>
> ## 2. This work is not the first to explore deductive reasoning in the multi-modal domain, please refer to compositional reasoning in VideoQA and ImageQA for more information. These task are not same as your task, but they explore programs to model the deductive reasoning in the multi-modal domain.
>
> We appreciate your suggestions about comparing with these well-established tasks (VideoQA and ImageQA). We will clarify the difference between these tasks and our ART ask. Then, we will update the complete task comparison to the next paper version.
>
> Referring to the format of the deductive reasoning task in the NLP field [1], we design the multi-modal deductive reasoning, ART. Both our task and the NLP deductive reasoning task [1] require the AI system to perform deductive reasoning under specific explicit rule constraints
>
> Based on this requirement, our ART task has three characteristics different from the VideoQA/ImageQA: (1) the explicit rule set is provided; (2) the rules used for each example are explicitly restricted; (3) the rule-based reasoning process needs to be provided by the AI system.
>
>
>
> ## 3. The over claim towards ``the first'' is not a big deal. My main concerns is the real-world application, can you provide a more applicable usage towards your model and task. For example, whether the rule set can be generated by LLMs? And how is the performance of your method when facing generated rule set? Moreover, whether your dataset can be further expanded by LLMs in the aforementioned approach?
>
> ### a) The over claim towards ``the first''.
>
> Thank you very much for your understanding and reminder! In the next paper version, we will modify the claim about our ART task in the multi-modal deductive reasoning field. Once again, thank you.
>
>
> ### b) My main concerns is the real-world application, can you provide a more applicable usage towards your model and task.
>
> We understand your concerns about the real-life application of our ART task. As described in the response to question 1, one potential application of our work is to adjudicate different types of cases in the judicial field, based, based on different legal provisions. Specifically, according to different legal provisions, the rule sets are defined. The AI system with deductive reasoning abilities can assist judicial decisions based on video evidence and rule sets. We will update our paper and add the real-world application description to the main body.
>
> ### c) For example, whether the rule set can be generated by LLMs? And how is the performance of your method when facing generated rule set?
>
> Generating the rule set using LLMs is an interesting idea, which is the next step of our research. Recently, we plan to contact the Microsoft company and use the GPT-4 to generate more rule sets. We plan to add the complete experiment results on the generated rule sets to the next version of our paper, and discuss the significance of the multimodal LLM in obtaining comprehensive rule sets.
>
> In real-world applications, the rule sets in certain fields are not suitable to be generated solely using LLMs. For example, in the field of judiciary, rules generated directly by LLMs often fail to align with actual legal provisions. Therefore, judicial rules must be formulated based on legal provisions and undergo meticulous manual review.
>
> ### d) Moreover, whether your dataset can be further expanded by LLMs in the aforementioned approach?
>
> This is also an interesting idea. In the future, we plan to generate more examples based on the GPT-4.
> The extension includes two aspects: (1) Modifying existing examples in our dataset to generate new ones; (2) Creating entirely new examples based on the generation capabilities of GPT-4.
>
> However, even with GPT-4, the AI system may not generate a 100% correct rule set. If so, we also need human annotators and verifiers to further annotate and verify the examples generated by the LLMs.
>
>
> [1] Sanyal S, Singh H, Ren X. Fairr: Faithful and robust deductive reasoning over natural language[J]. arXiv preprint arXiv:2203.10261, 2022.

---

### Official Review · Reviewer_XipY · 2023-08-03

**Soundness:** 4

**Excitement:**

3: Ambivalent: It has merits (e.g., it reports state-of-the-art results, the idea is nice), but there are key weaknesses (e.g., it describes incremental work), and it can significantly benefit from another round of revision. However, I won't object to accepting it if my co-reviewers champion it.

**Paper Topic And Main Contributions:**

The paper proposes a new task that deduces future events based on rules and visual information. It constructs a dataset and a baseline model for the new task. Experimental results demonstrate the effectiveness of the proposed model.

**Reasons To Accept:**

- The proposed task is interesting and has the potential to promote rule-based reasoning in the multimodal domain.
- The paper is well-written, offering a comprehensive task definition and clear formulation.

**Reasons To Reject:**

- The setting of confusing items in the candidate events is a critical aspect of the dataset quality. However, the paper lacks a detailed description of this aspect. The provided cases in the paper seem to contain relatively unconfusing incorrect candidates, that is, it seems easy to distinguish the correct candidate from the incorrect ones based on relevance or similarity to the rules. Addressing this concern would strengthen the paper's dataset construction.
- The paper lacks analyses on dataset bias. First, it remains unclear how crucial each modality (image or text) is to the new task. Evaluating the performance of models that solely rely on image or text inputs would provide insights into the importance of each modality. Second, there is a potential issue of models exploiting the relevance or similarity between candidates and rules as a shortcut, rather than genuinely reasoning through the problem. Evaluating and discussing this dataset bias is essential to ensure the dataset quality.
- It would be better to incorporate potential applications or downstream tasks of the new task, which may enhance the significance of the new task.

**Reproducibility:**

4: Could mostly reproduce the results, but there may be some variation because of sample variance or minor variations in their interpretation of the protocol or method.

**Reviewer Confidence:**

3: Pretty sure, but there's a chance I missed something. Although I have a good feel for this area in general, I did not carefully check the paper's details, e.g., the math, experimental design, or novelty.

---

> ### Author Rebuttal · Authors · 2023-08-29
>
> ## Rebuttal to Reviewer XipY
>
> We appreciate your valuable suggestions. Thank you very much! We will carefully respond to the comments item by item.
>
> ## 1. The setting of confusing items in the candidate events is a critical aspect of the dataset quality. However, the paper lacks a detailed description of this aspect. The provided cases in the paper seem to contain relatively unconfusing incorrect candidates, that is, it seems easy to distinguish the correct candidate from the incorrect ones based on relevance or similarity to the rules. Addressing this concern would strengthen the paper's dataset construction.
>
> ### a) The setting of confusing items in the candidate events is a critical aspect of the dataset quality. However, the paper lacks a detailed description of this aspect.
>
> We understand your concerns and are sorry for the incomplete description of the confusing item construction. We will describe the construction process in detail.
>
> Before generating a confusing item, a correct event is constructed. Then, we inject an error into the correct event.
> There are several ways of injecting errors to choose:
>
> 1. Change the correct appearance and clothing description into the error one, like the description of another character in the video.
>
> 2. Replace the correct rule-based reasoning process with a logically inconsistent rule-based reasoning process. Then, change the correct event into the confusing event corresponding to the logically inconsistent reasoning process.
>
> 3. Replace the correct rule-based reasoning process with another logically consistent reasoning process that contradicts the video details. Then, change the correct event into the confusing event corresponding to the new reasoning process.
>
> 4. Inject some description contradicting the video details into the correct item and change it into a confusing item.
>
> 5. Inject some description contradicted with the common sense into the correct item and change it into a confusing item.
>
> 6. We also encourage the annotators to choose other error injection methods specific to the particular videos, such as contradicting the scenes in the videos.
>
> The verifiers check all the confusing items carefully, to ensure the quality of these items.
>
>
> ### b) The provided cases in the paper seem to contain relatively unconfusing incorrect candidates, that is, it seems easy to distinguish the correct candidate from the incorrect ones based on relevance or similarity to the rules. Addressing this concern would strengthen the paper's dataset construction.
>
> We understand your concerns about that there are some unconfusing items in our cases. We randomly select the cases from our dataset to present our model performance.
> These individual cases cannot effectively represent our entire dataset from the perspective of the candidate event difficulty.
> Our dataset includes examples and candidate events of various difficulty levels. Many candidate events are difficult to distinguish, which requires a combination of selecting appropriate rules and considering video details to make accurate inferences.
> We apologize for the incomplete selection of visual cases.
> We will add more cases to the main body of the next paper version, to completely show examples of different difficulty levels.
>
> In addition, to provide the testbed for both the early-stage models (poor performance) and later-stage models (good performance), there are both easy-to-judge and difficult-to-judge candidate events in our dataset. From the perspective of model accuracy, the current dataset has a moderate level of difficulty. It distinguishes the inference capabilities of the current-stage models while leaving room for future research advancements.
> When the models reach the accuracy limit on our dataset, we plan to construct a more challenging dataset to further promote the development of this field.
>
>
> ## 2. The paper lacks analyses on dataset bias. First, it remains unclear how crucial each modality (image or text) is to the new task. Evaluating the performance of models that solely rely on image or text inputs would provide insights into the importance of each modality. Second, there is a potential issue of models exploiting the relevance or similarity between candidates and rules as a shortcut, rather than genuinely reasoning through the problem. Evaluating and discussing this dataset bias is essential to ensure the dataset quality.
>
>
>
> ### a) The paper lacks analyses on dataset bias. First, it remains unclear how crucial each modality (image or text) is to the new task. Evaluating the performance of models that solely rely on image or text inputs would provide insights into the importance of each modality.
>
> Thank you for the reminder! We will complete the analysis of each modality function in the model reasoning process:
>
> ***Analysis of the function of the input visual information.*** We black out all the videos in our dataset. Then, we train and evaluate our model ARTNet with the text and these masked videos, to verify the function of the input videos. The experiment results in the real-life scene are shown in the following:
>
> | Methods | Event_ACC | Exp_ACC |
> | ------ | ------ | ------ |
> | ARTNet | 42.3 | 41.0 |
> | ARTNet (with masked videos) | 27.3 | 24.1 |
>
> Visual information serves as the foundation for our task's reasoning process. After blacking out the input videos, the accuracy of our ARTNet model has significantly declined. This proves the importance of visual information in the inference process of the AI system.
>
> ***Analysis of the function of the input text information.*** The ART task requires the AI system to judge the correctness of all candidate events and select the correct one. Thus, it is necessary for the AI system to get the input of the candidate event text. Thus, when verifying the effectiveness of the input text, we only remove the participation of the textual rule set, during the training and testing process of our ARTNet. The experiment results in the real-life scene are shown in the following:
>
> | Methods | Event_ACC | Exp_ACC |
> | ------ | ------ | ------ |
> | ARTNet | 42.3 | 41.0 |
> | ARTNet (without rule sets) | 36.2 | - |
>
> It is worth noting that without the rule sets, the rule-based explanations cannot be inferred by our ARTNet, whose accuracy (Exp_ACC) cannot be calculated.
>
> From the experiment results, it can be observed that the ARTNet performance drops significantly without rule sets. It proves that the importance of the textual rule set in the reasoning process of the AI system. The complete experiment results will be added to the next version of our paper.
>
>
> ### b) Second, there is a potential issue of models exploiting the relevance or similarity between candidates and rules as a shortcut, rather than genuinely reasoning through the problem. Evaluating and discussing this dataset bias is essential to ensure the dataset quality.
>
> We understand your concerns about the shortcut of only applying the candidates and rules in the reasoning process. It is necessary to prove the importance of visual information in the reasoning process. As shown in the response to question 2-a), the performance of our ARTNet drops significantly, without the visual information. This is because crucial information for determining the correctness of candidate events, such as the appearance and clothing of target characters, initial actions of characters, etc., must be obtained from the videos. The absence of visual information will result in an incomplete reasoning basis for the AI system.
> We will add more experiment results and analysis about the genuine reasoning of models to the next version of our paper.
>
>
> ## 3. It would be better to incorporate potential applications or downstream tasks of the new task, which may enhance the significance of the new task.
>
> We appreciate the valuable suggestions! There are numerous potential applications for our work. One significant application is in legal judgment. Specifically, in the judicial scenario, we can predefine rule sets based on various legal provisions, such as environmental protection laws, civil laws, and more. In specific cases, an AI system with deductive reasoning abilities can effectively combine video evidence with rule sets to facilitate the judicial process.
>
> In the next version of our paper, we will provide a more detailed description of the applications of our ART task in the main body of the paper.

---

### Official Review · Reviewer_Gibb · 2023-08-13

**Soundness:** 4

**Excitement:**

4: Strong: This paper deepens the understanding of some phenomenon or lowers the barriers to an existing research direction.

**Paper Topic And Main Contributions:**

This work proposes a new ART task (rule based future-inference deduction task) under the multi-modal setting. It collects a dataset called Video-ART consisting of 23,895 examples with rule set and reasoning process annotations. Based on it, a strong baseline model ARTNet is also developed with three modules: 1) knowledge-guided target perception with encoder-decoder transformers, 2) rule-based graph reasoning network, and 3) reasoning path review to decide the correct future event.

**Questions For The Authors:**

A. Is it possible to provide the IAA for the data collection process? I saw the details in the appendix to ensure data quality, but it would be great to know a bit more statistically.

B. How did the authors get all rules (i.e. rule storage)? Is it from the whole dataset (i.e. dataset dependent)?


**Reasons To Accept:**

The paper formulates the ART task as an early exploration of deductive reasoning in the multi-modal domain. This work provides a large and rich annotated dataset which could benefit the research related to this topic. Also, authors develop a well-designed baseline towards better understanding the task. Overall it is complete and solid from my point of view.

**Reasons To Reject:**

The proposed ART task is close to the well-established video inference task but with more details in commonsense rules and reasoning process. Though it is an interesting direction, based on the restrictions, it might be a bit hard to scale for future researchers.

**Reproducibility:**

4: Could mostly reproduce the results, but there may be some variation because of sample variance or minor variations in their interpretation of the protocol or method.

**Reviewer Confidence:**

3: Pretty sure, but there's a chance I missed something. Although I have a good feel for this area in general, I did not carefully check the paper's details, e.g., the math, experimental design, or novelty.

**Typos Grammar Style And Presentation Improvements:**

A. L207: Would like to know a bit of copyright related discussions for the 3,983 manually collected video clips (i.e. are those 80 American movies free to use and distribute without license issues?)

B. L243, L256: It is minor, but for more broader audiences, wording like `from the top 50 universities in the world` sounds a bit not appropriate.

---

> ### Author Rebuttal · Authors · 2023-08-29
>
> ## Rebuttal to Reviewer Gibb
>
> Thank you for your patient feedback! We are delighted to offer detailed responses and make comprehensive revisions to our paper based on these valuable suggestions.
>
> ## 1. The proposed ART task is close to the well-established video inference task but with more details in commonsense rules and reasoning process. Though it is an interesting direction, based on the restrictions, it might be a bit hard to scale for future researchers.
>
> We understand your concerns about how to follow our work for future researchers. We will provide a detailed and careful response to the concerns:
>
> ***Proposing new models for the ART task.*** We propose a strong baseline, ARTNet, and evaluate the performance of our model and several state-of-the-arts on our Video-ART dataset. In the future, we plan to release the code of our ARTNet and other state-of-the-arts, for the convenience of future researchers.
> As claimed in our paper, our ARTNet is a strong baseline for the ART task.
> The current model's accuracy is far from reaching its upper limit. Perhaps there are better methods for rule utilization and video detail analysis that can enhance the model's capabilities. We hope that based on our ARTNet, more researchers will propose higher-accuracy models.
>
> ***Proposing new datasets for the ART task*** Compared with the well-established video inference task, our CORE task requires commonsense rule annotations, which brings difficulty to the annotation. However, deductive reasoning is a basic recognition ability of humans, detailed in lines 55-57 of our paper. In practice, many annotators with certain reasoning abilities can annotate the commonsense rule labels and other deductive reasoning annotations effectively. With a sufficient number of such annotators or ample time, the deductive reasoning dataset can be substantially expanded. In the future, if other researchers would like to expand our dataset, we will provide enough help with training annotators, quality control checks, and other related measures.
>
> ***Contribution to other related tasks*** We annotate various types of commonsense knowledge, including sentiment and actions. This allows our work to provide a solid data foundation for tasks such as video emotion recognition or action recognition.
>
> ## 2. Is it possible to provide the IAA for the data collection process? I saw the details in the appendix to ensure data quality, but it would be great to know a bit more statistically.
>
> Yes, we are happy to introduce the annotation process in detail. We guarantee the dataset quality at each stage of its construction:
>
> ***Data Collection***
> 1. ***Personnel Training.***
> We provide separate centralized training for data collectors and verifiers, informing them of the standards for data collection and verification. Specifically, there are many collection standards to follow. For example, the collected videos need to be sufficiently clear and information-rich to avoid prolonged periods of low-information static scenarios. For the verification standards, the verifiers need to check each collected video one by one to ensure the quality, based on the collection standards.
>
> 2. ***Double Verifications.***
> There are double verifications for the collected videos. Specifically, except for the verification of the verifiers, the trainers are responsible for conducting random spot checks on 30% of data that has already undergone comprehensive verification. We hope the double verification can ensure the quality of the collected videos.
>
> ***Data Annotation***
> 1. ***Multiple Training Sessions.***
> The annotators are trained for several times. Before annotation, we provide centralized training for all annotators. After training, we conduct annotation tests for them and select annotators with high accuracy for the subsequent annotation. During the annotation process, if new issues are identified by verifiers, we will conduct centralized retraining for the annotators and perform comprehensive revisions for annotated examples. For example, different annotators have different standards for "blouse" and "shirt". The trainers find reference images as standards to retrain the annotators, and require them to perform rework.
>
> 2. ***Double Verifications.***
> Similar to the process of data collection, there are double verifications in the annotation process. Firstly, the verifiers check the annotated examples one by one. Then, the trainers of the verifiers randomly sample 30% of the labeled examples, and check them carefully to further verify the annotation quality.
>
> With rigorous personnel training, we found that the agreement rate between annotators and verifiers reached 95% for the same annotation test. We will add these annotation details to our paper to completely introduce our dataset. Thank you for the valuable suggestions!
>
>
> ## 3. How did the authors get all rules (i.e. rule storage)? Is it from the whole dataset (i.e. dataset dependent)?
>
> ### a) How did the authors get all rules (i.e. rule storage)?
>
> We construct the rule storage based on the rule set of the input example.
> Specifically, following the previous deductive reasoning task [1] in the NLP field, each example is accompanied by an independent rule set. We transform this rule set into many action chains, detailed in lines 303-319 of our paper. When analyzing each input example, the rule storage is constructed based on the action chains corresponding to the given example.
>
> We are sorry for the unclear description of the rule storage construction. We will add the detailed construction process into the main body of our paper.
>
> ### b) Is it from the whole dataset (i.e. dataset dependent)?
>
> No, the rule storage is constructed based on the rule set of the input example, detailed in question 3-a).
>
> ## 4. L207: Would like to know a bit of copyright related discussions for the 3,983 manually collected video clips (i.e. are those 80 American movies free to use and distribute without license issues?)
>
> About the copyright of our dataset, we plan to follow the protection method of the widely known dataset, ImageNet [2], whose examples are collected from the internet. Specifically, we provide the thumbnail frames of the videos and a copyright infringement takedown notice. Only after signing an agreement and guaranteeing that they will not commercially use our dataset, we will provide the original video links, as well as the start and end timestamps of the used video clips.
>
> ## 5. L243, L256: It is minor, but for more broader audiences, wording like from the top 50 universities in the world sounds a bit not appropriate.
>
> Thank you for the valuable reminder! We are sorry for the inappropriate description. We will change it to "the undergraduate and graduate students with strong reasoning abilities" in the next paper version. Thank you!
>
> [1] Sanyal S, Singh H, Ren X. Fairr: Faithful and robust deductive reasoning over natural language[J]. arXiv preprint arXiv:2203.10261, 2022.
>
> [2] Deng J, Dong W, Socher R, et al. Imagenet: A large-scale hierarchical image database[C]//2009 IEEE conference on computer vision and pattern recognition. Ieee, 2009: 248-255.

---

### Meta-Review · Area_Chair_vTmz · 2023-09-18

**Recommendation:** 5

**Metareview:**

This paper proposes a new ART task (rule based future-inference deduction task) to explore the deductive reasoning in multi-modal domain. Moreover, this paper construct a large-scale dataset to promote multi-modal deductive reasoning research, which consists of 23, 895 examples where dense annotations including the rule set, reasoning processes, and auxiliary commonsense knowledge are provided. Based on it, a strong baseline model ARTNet is also developed with three modules: 1) knowledge-guided target perception with encoder-decoder transformers, 2) rule-based graph reasoning network, and 3) reasoning path review to decide the correct future event. Experimental results demonstrate the effectiveness of the proposed model.
All reviewers rate soundness and excitement highly. In summary, the paper formulates the ART task as an early exploration of deductive reasoning in the multi-modal domain. This work provides a large and rich annotated dataset which could benefit the research related to this topic. Also, authors develop a well-designed baseline towards better understanding the task. The proposed task is interesting and has the potential to promote rule-based reasoning in the multimodal domain. The paper is well-written, offering a comprehensive task definition and clear formulation. This paper has great potential for accept to main conference.

---

### Decision · Program_Chairs · 2023-10-07

**Decision:**

Accept-Main

**Comment:**

This paper proposes a new ART task (rule based future-inference deduction task) to explore the deductive reasoning in multi-modal domain. Moreover, this paper construct a large-scale dataset to promote multi-modal deductive reasoning research, which consists of 23, 895 examples where dense annotations including the rule set, reasoning processes, and auxiliary commonsense knowledge are provided. Based on it, a strong baseline model ARTNet is also developed with three modules: 1) knowledge-guided target perception with encoder-decoder transformers, 2) rule-based graph reasoning network, and 3) reasoning path review to decide the correct future event. Experimental results demonstrate the effectiveness of the proposed model.
All reviewers rate soundness and excitement highly. In summary, the paper formulates the ART task as an early exploration of deductive reasoning in the multi-modal domain. This work provides a large and rich annotated dataset which could benefit the research related to this topic. Also, authors develop a well-designed baseline towards better understanding the task. The proposed task is interesting and has the potential to promote rule-based reasoning in the multimodal domain. The paper is well-written, offering a comprehensive task definition and clear formulation. This paper has great potential for accept to main conference.